# Determinants of Knowledge and Attitude towards Breastfeeding in Rural Pregnant Women Using Validated Instruments in Ethiopia

**DOI:** 10.3390/ijerph18157930

**Published:** 2021-07-27

**Authors:** Misra Abdulahi, Atle Fretheim, Alemayehu Argaw, Jeanette H. Magnus

**Affiliations:** 1Department of Population and Family Health, Jimma University, Jimma 378, Ethiopia; yemariamwork2@gmail.com; 2Department of Community Medicine and Global Health, University of Oslo, 0316 Oslo, Norway; 3Faculty of Health Sciences, Oslo Metropolitan University, 0130 Oslo, Norway; Atle.Fretheim@fhi.no; 4Centre for Informed Health Choices, Norwegian Institute of Public Health, 0473 Oslo, Norway; 5Department of Food Technology, Safety and Health, Faculty of Bioscience Engineering, Ghent University, B-9000 Ghent, Belgium; 6Faculty of Medicine, University of Oslo, 0316 Oslo, Norway; j.h.magnus@medisin.uio.no; 7Department of Global Community Health and Behavioral Sciences, Tulane School of Public Health and Tropical Medicine, New Orleans, LA 70112, USA

**Keywords:** attitude, breastfeeding, determinants, knowledge, rural Ethiopia

## Abstract

Understanding the underlying determinants of maternal knowledge and attitude towards breastfeeding guides the development of context-specific interventions to improve breastfeeding practices. This study aimed to assess the level and determinants of breastfeeding knowledge and attitude using validated instruments in pregnant women in rural Ethiopia. In total, 468 pregnant women were interviewed using the Afan Oromo versions of the Breastfeeding Knowledge Questionnaire (BFKQ-AO) and the Iowa Infant Feeding Attitude Scale (IIFAS-AO). We standardized the breastfeeding knowledge and attitude scores and fitted multiple linear regression models to identify the determinants of knowledge and attitude. 52.4% of the women had adequate knowledge, while 60.9% of the women had a neutral attitude towards breastfeeding. In a multiple linear regression model, maternal occupation was the only predictor of the BFKQ-AO score (0.56SD; 95%CI, 1.28, 4.59SD; *p* = 0.009). Age (0.57SD; 95%CI, 0.24, 0.90SD; *p* = 0.001), parity (−0.24SD; 95%CI, −0.47, −0.02SD; *p* = 0.034), antenatal care visits (0.41SD; 95%CI, 0.07, 0.74SD; *p* = 0.017) and the BFKQ-AO score (0.08SD; 95% CI, 0.06, 0.09SD; *p* < 0.000) were predictors of the IIFAS-AO score. Nearly half of the respondents had inadequate knowledge and most women had a neutral attitude towards breastfeeding. Policymakers and managers could address these factors when planning educational interventions to improve breastfeeding practices.

## 1. Introduction

Exclusive breastfeeding (EBF) of infants during the first six months of life is recommended [1], given the several benefits that have been identified for both mother and infant [2,3,4,5,6,7,8,9,10,11,12]. Despite the presence of global recommendations, the practice of EBF substantially lags behind the level recommended by the World Health Organization, especially in lower- and middle-income countries where suboptimal infant feeding is prevalent [13]. In Ethiopia, early initiation of breastfeeding is at 78%, while 58% of children under 6 months old are exclusively breastfed [14]. Although the current rate of early initiation and EBF are high compared to the overall global rate, both are below the 2020 national targets of 90% and 72%, respectively [15]. Moreover, after birth, the EBF rate in Ethiopia declines rapidly from 74% between 0–1 month to 36% at 4–5 months [14]. Further, although the rate of EBF increased from 55% in 2000 to 60% in 2016, this increase was neither substantial nor forecasting the reach of the national 2020 targets.

Understanding the levels and underlying determinants of maternal knowledge and attitude towards breastfeeding guides the development of context-specific interventions aimed at increasing the rates of optimal breastfeeding practices. Large international studies undertaken to estimate the prevalence and determinants of EBF practice have identified different sociodemographic and psychosocial factors, such as a mother’s knowledge and attitude [16,17,18]. The knowledge and attitude of women towards breastfeeding are known to affect their decision to breastfeed [19]. Women who do not have adequate knowledge do not practice breastfeeding, as they do not understand the benefits and importance of doing so [20]. Furthermore, a positive maternal attitude toward breastfeeding is a stronger predictor of breastfeeding initiation and a longer duration than sociodemographic factors [21,22,23,24,25,26,27]. Because women’s knowledge and attitude are modifiable factors associated with breastfeeding outcomes, they are important individual-level variables that should be targeted for behavioural change.

In Ethiopia, existing studies have attempted to recognize the critical role played by maternal knowledge and attitude as predictors of breastfeeding practices [28,29,30,31,32,33,34,35,36,37]. Maternal knowledge was identified as a predictor in a few of these studies [30,33,34,35,36,37]. In addition, the attitude towards breastfeeding has been assessed, but a significant association was not found [29,32,33,36]. Surprisingly, the effect of maternal knowledge and attitude remain unclear, as these studies vary greatly in their depth, in terms of the quantity and content of the questions used. The knowledge questions are superficial and lack some important domains, such as breastmilk expression, management of breast problems and practical aspects of breastfeeding. Moreover, they have used non-validated tools to report varied factors across different settings. Due to the lack of validated instruments in Ethiopia, there is a dearth of data on a range of factors known to influence maternal knowledge and attitude towards breastfeeding, rendering the generalizability of previous studies on this issue problematic.

The strength of the evidence to support interventions depends on the quality and sensitivity of instruments used to measure outcomes [38], and the quality of a measuring instrument is related to its reliability and validity [39]. Valid and reliable instruments to measure knowledge and attitude assists policymakers, researchers and practitioners develop strategies on how to promote breastfeeding [38], by identifying the levels and predictors of maternal knowledge and attitude towards breastfeeding. As such, interventions that aim to improve psychosocial factors, such as knowledge and attitude, that influence the decision-making process regarding breastfeeding, require instruments that are objective, reliable and valid to measure these outcomes [38]. A systematic review of maternal knowledge about breastfeeding and attitude toward infant feeding suggests that attitude towards infant feeding should be evaluated using the standard and validated instrument of the Iowa Infant Feeding Attitude Scale (IIFAS) [38]. Thus, we opted to use IIFAS to assess attitude. For the knowledge assessing questionnaire, we conducted a literature search to find a valid and reliable instrument to assess maternal knowledge about breastfeeding, and we found two instruments: the Breastfeeding Knowledge Questionnaire that was developed in America [40], and a questionnaire assessing knowledge of breastfeeding that was developed in Malaysia [41]. We preferred to use the latter, as it contains domains of breastfeeding that were not included in the Breastfeeding Knowledge Questionnaire, such as colostrum, breastmilk expression, management of breast problem and practical aspects of breastfeeding.

With the goal of improving breastfeeding rates in rural Ethiopia, we conducted a randomized controlled trial in which we provided pregnant women with a breastfeeding education and support intervention (BFESI), that comprised of prenatal breastfeeding education and postnatal peer support [42]. As one of the aims of the research project was to assess changes in the level of knowledge and attitude towards breastfeeding, in the current study, we sought to measure the baseline level and identify predictors of knowledge and attitude towards breastfeeding among women who enrolled into the BFESI trial. This study will provide new insight into the factors that influence knowledge and attitude towards breastfeeding, using instruments that were validated in an Ethiopian context.

## 2. Materials and Methods

### 2.1. Design

This is a cross-sectional study that used data that were collected between May and August 2017 for the baseline survey of the BFESI. The protocol is published elsewhere [42]. The protocol was approved by the institutional review boards of Jimma University and Oromia regional health bureau.

### 2.2. Setting

The study was conducted in Mana district, one of the twenty-one districts found in Jimma zone, Oromia region, Southwest Ethiopia. The zone is located 368 km from Addis Ababa and 22 km from Jimma town. Agriculture is the main form of livelihood in the study community, with coffee accounting for 80% of the main crops produced in the area. According to the Manna District Health Office, the district had a total population of 197,911 in 2019. The district has a total of 26 kebeles (the smallest administrative unit in Ethiopia, 1 urban and 25 rural). It has 7 health centres, 26 health posts, 11 private clinics and 3 private pharmacies. The district has 68 health extension workers and 121 healthcare providers of different professions. The area is known for its shortage of human resources for health, as evidenced by the ratio of 121 health workers to 1000 people, i.e., 0.2, which is far below the national target of 1.6 by 2020 [15] and the international target of 4.5 [43], indicating the area has inadequate healthcare services.

### 2.3. Sample

We recruited 468 pregnant women using the Ethiopian community health system: the Health Extension Workers antenatal logbook and Women Development Army Leaders. All pregnant women who were identified from the antenatal care logbook were invited to a meeting at the health post, where the nature and purpose of the trial and eligibility criteria were explained, including their right to withdraw from the study at any time. They were allowed to ask questions and relevant information was provided accordingly. Then, according to their literacy status, written informed consent was secured from participants by asking them to provide their fingerprint or signature. To minimize the chance of missing pregnant women, Women Development Army Leaders were involved in the identification of pregnant women who were not registered on the antenatal care logbook and consent was obtained through a home visit following the above procedures. Participants comprised of healthy, pregnant women in their 2nd or 3rd trimester without severe health complications, including any psychiatric illness, who provided consent to participate in the trial and stated that they had no plan to move from the study area before completion of the BFESI trial.

### 2.4. Data Collection

Ten BSc nurses who had experience in data collection and were fluent in the local language completed a 2 day training on the main objectives of the study and how to standardize the data collection method. All the required supervision was provided by trained supervisors during the training session and on the field during data collection. Data were collected using a structured face-to-face interview after obtaining written consent from each study participant. Maternal knowledge and attitude towards breastfeeding were assessed using the local language versions of the Breastfeeding Knowledge Questionnaire-Afan Oromo (BFKQ-AO) and the Iowa Infant Feeding Attitude Scale-Afan Oromo (IIFAS-AO), which were locally adapted and validated in a similar rural population [44]. Furthermore, data on potential determinants of knowledge and attitude were gathered. Sociodemographic factors, including maternal age, level of education, number of children, household wealth and food security status were collected in addition to maternal factors, including parity, antenatal care (ANC) visits, past obstetric and breastfeeding history.

### 2.5. Measurement

The BFKQ-AO consists of 34 items asking about various optimal breastfeeding practices, with responses coded as correct or incorrect. We decided to use a cut-off above or below the median to dichotomize the knowledge level. The breastfeeding knowledge questionnaire adopted from Malaysia had not operationalized an optimal knowledge level [41]. Accordingly, all women who scored ≥ the median in the knowledge test were considered as having a high level of knowledge, and those scoring below the median were considered as having a low level of knowledge. The IIFAS-AO consists of 17 items with a 5-point Likert scale, rating maternal attitude towards breastfeeding, and the total scores range from 17 to 85 with a higher score reflecting a positive attitude. Attitude toward breastfeeding was categorized as follows: (1) positive to breastfeeding (IIFAS score 70–85), (2) neutral (IIFAS score 49–69) and (3) positive to formula feeding (IIFAS score 17–48) [24].

### 2.6. Data Analysis

Double data entry were performed using EpiData (version 3.1), and all statistical analyses were completed using Stata version 13.1 (StataCorp LLC: College Station, TX, USA). Data were summarized using frequencies and percentages. In the first stage of the analysis, we evaluated bivariate associations between potential predictors and the study outcomes, with breastfeeding knowledge and attitude scores to determine candidate predictors for the subsequent multiple linear regression models. In the second stage, based on the result of bivariate associations, we fit multiple linear regression models assessing the independent predictors of breastfeeding knowledge and attitude scores. We applied a robust variance estimation to take into account the clustering of subjects by study sub-districts. The knowledge and attitude scores were standardized based on the distribution of our data, and the results are expressed as regression coefficients with 95% Confidence Intervals (CIs). Models were evaluated for potential multi-collinearity using the variance inflation factor, with values less than ten considered acceptable. Model goodness of fit was assessed using adjusted R2 values. All tests were two-tailed and a statistically significant association was considered at a *p*-value < 0.05.

## 3. Results

The characteristics of the 468 pregnant women are presented in Table 1. Most women were in the age group of 20–34 (83.3%), illiterate (74.6%), housewives/farmers (93.8%) and lived in a food-insecure household (58.9%). Three hundred and eighty-two (81.6%) of the enrolled women were multiparous, of which 96.8% had a history of breastfeeding. The majority (88%) of the women had at least one ANC visit, but only 7.28% of them had the recommended ≥4 antenatal care visits (Table 1).

### 3.1. Levels of Breastfeeding Knowledge and Attitude

The mean ± SD of the overall IIFAS score in our sample was 65.7 ± 7.6 points with a range between 36 and 85 points. The majority (60.9%) of the women had a neutral attitude towards breastfeeding, whereas 36.9% of the participants had strongly positive attitude toward breastfeeding. Two hundred and forty-five (52.4%) of the womenwomen had a high level of knowledge, while 223 (47.6%) had a low level of knowledge. Aspects of knowledge domains in which women scored high were the advantage to baby, the advantage to mother, effective feeding, duration of feeding and the practical aspect of breastfeeding. However, they scored low on colostrum, breast milk expression, problems with breastfeeding and breast engorgement domains (data not shown).

### 3.2. Predictors of Breastfeeding Knowledge and Attitude

From the variables included in the multiple linear regression model, i.e., age, education, employment, wealth tertile, number of children and ANC visit, only maternal occupation significantly predicted breastfeeding knowledge score (Table 2). Compared to women who were housewives, women who involved in small trades and private employment had a significantly higher breastfeeding knowledge score (β: 0.56 SD; 95% CI, 0.14, 0.97 SD; *p* = 0.009). The variables that independently predicted maternal attitude towards breastfeeding included maternal age, education, employment and parity, household wealth status and ANC visit (Table 3). Older women, as compared to their younger counterparts (<20 years) (β: 0.57 SD; 95% CI, 0.24, 0.90 SD; *p* = 0.001), and primiparous women, as compared to multiparous (β: −0.24 SD; 95%CI, −0.47, −0.02SD; *p* = 0.034), had a significantly higher IIFAS score. Women who attended at least four ANC visits were found to have a significantly higher attitude towards breastfeeding than those who had no ANC visit (β: 0.41 SD; 95% CI, (0.07, 0.74 SD; *p* = 0.017). We also demonstrate a statistically significant positive association between maternal BFKQ-AO and IIFAS scores (β: 0.33 SD; 95% CI, 0.25, 0.41 SD; *p* < 0.000).

## 4. Discussion

Initiation, duration and exclusivity of breastfeeding is a choice made by all women, but is highly affected by various factors, including knowledge about and attitude towards the benefits of breastfeeding. Despite Ethiopian attempts at increasing the rate of early initiation and duration of exclusive breastfeeding, its success has been limited. To the best of our knowledge, this is the first study assessing predictors of knowledge and attitude towards breastfeeding using validated instruments in a local language in Ethiopia. This community-based study revealed that half of the women had adequate knowledge about breastfeeding. We also found that knowledge was associated only with maternal occupation, while the women’s attitude towards breastfeeding was associated with maternal age, parity, antenatal care visits and their overall knowledge about breastfeeding.

By definition, a significant proportion of pregnant rural women had a low level of breastfeeding knowledge in this study. Interventions aimed at improving breastfeeding knowledge are important in efforts to encourage women to breastfeed. For Ethiopia, this might be imperative to reach the targets set in the national health sector plan [15]. Particularly, questions relating to aspects of knowledge that scored very low, such as colostrum, breastmilk expression, issues related to breastfeeding problems and breast engorgement, need to be emphasized during breastfeeding promotion. In Ethiopia, although extensive research has been carried out on the knowledge of women on breastfeeding [30,33,34,35,36], no single study exists which applies a validated instrument, and comparing our findings with these studies is unrealistic.

A positive attitude toward breastfeeding is a stronger predictor of breastfeeding initiation and duration when compared to sociodemographic factors [21,22,23,24,25,26,27]. In our study, the mean IIFAS-AO score was within a range that reflected neutral attitude towards breastfeeding [24]. The findings from the current study corroborate similar findings in the literature, as a neutral attitude towards breastfeeding has been reported across a diverse group of countries, including China and Australia [16,45], Spain [46], Canada [47], Japan [48] and Scotland [22,49]. While it is possible that the finding of a neutral attitude could be one of the major reasons for a high level of breastfeeding initiation, the low level of EBF in Ethiopia needs further study. In Ethiopia, none of the studies that assessed women’s attitude towards breastfeeding [29,32,33,36] used IIFAS, and much uncertainty exists around the relationship between maternal attitude and breastfeeding practice. Our study provides a new insight into the importance of considering neutral attitude.

In the present study, women who were involved in small trades and private employment had a higher knowledge about breastfeeding. This is similar to a study in China [17], which reported a greater knowledge about breastfeeding among employed women. This could be expected, as employment is associated with education and increased general knowledge.

Earlier studies underscore that attitude influence behaviour; therefore, knowing something about a person’s attitude can help predict behaviour in many contexts [50]. In this study, we found that younger women had higher attitude scores. In contrast, previous studies in China [16], Taiwan [51], Singapore [52] and Romania [53] showed that older women had more positive attitude toward breastfeeding. The direct relationship between a higher IIFAS-AO score (positive attitude) and increased age is likely linked to prior experience with breastfeeding. Breastfeeding is in many aspects a learned behaviour.

Maternal education is associated with favorable attitude toward breastfeeding, as highly educated women were more aware of the breastfeeding benefits. Although not statistically significant in the current study, women with a primary school education had a lower attitude score. Nevertheless, previous studies have shown that higher IIFAS scores were positively associated with a higher educational level in Lebanon [54], Ireland [55], China [17], Singapore [52] and Spain [46].

Studies report mixed results regarding the association between IIFAS score and occupation. A study from Ireland found that full or part-time employed women had more positive attitude toward breastfeeding compared to housewives [55], while a study among Chinese women reported no significant difference between infant feeding attitude and working status [16]. However, in our study of rural women, we found that those that were involved in small trades and private employment had a more negative attitude toward breastfeeding than homemakers did, but this difference was not statistically significant. It is conceivable that women who work may suffer from a lack of time, fatigue and experience breastfeeding as exhausting, leading to negative attitude towards breastfeeding.

Higher maternal IIFAS score is associated with higher family annual income in Lebanon [54], Taiwan [51], Singapore [52], Ireland [55] and Spain [46]. In our study, we generated a household asset score/wealth index, as household income does not measure the value of non-monetary items, particularly in a low-income context. Nevertheless, we did not find that the IIFAS-AO score was associated with the socioeconomic status of the household in our rural setting.

Multiparous women are expected to have a positive attitude towards breastfeeding, due to their prior experience, anticipated maternal confidence and ability to solve feeding problems. However, unlike findings from the Infant Feeding Survey, where multiparous women had more favourable attitude to breastfeeding [17], multiparous women in the current study were more negative towards breastfeeding. One can speculate that Ethiopian rural women with many children are taxed by household chores, high infant mortality and morbidity, food insecurity and limited time to breastfeed, all leading to negative attitude. On the other hand, women with fewer children might have more time, and may be more concerned with their newborn.

Pregnant women with ≥4 antenatal care visits at the time of the baseline survey in our study had higher IIFAS-AO scores. A possible explanation might be that these women had increased exposure to prenatal breastfeeding education/counselling. Moreover, our respondents with high levels of breastfeeding knowledge also had a higher attitude score, in line with a previous study from Finland [56].

### Limitations

Our study has some limitations. Firstly, this is a cross-sectional study design, and thus we cannot establish causal relationships. Our findings are associations that may or may not reflect cause and effect. Secondly, the majority of our rural Ethiopian participants had limited schooling, reducing the ability to detect a potentially statistically significant association between knowledge and attitude towards breastfeeding and educational status. The women in this study were demographically homogeneous, and the findings may not be applicable across different regions of the country. Additional research is needed to assess the relationship between IIFAS-AO and BFKQ-AO, as well as breastfeeding initiation and duration in urban Ethiopian settings.

## 5. Conclusions

A significant proportion of the rural pregnant women had inadequate knowledge about breastfeeding and a neutral attitude towards breastfeeding. The occupation of the women was identified as an independent predictor of breastfeeding knowledge, whereas age, parity, antenatal care visit and knowledge scores (BFKQ-AO) were predictors of breastfeeding attitude. Thus, policymakers and managers may address these factors when planning educational interventions on breastfeeding to improve knowledge and attitude, thereby advancing breastfeeding practices in rural communities.

## Figures and Tables

**Table 1 ijerph-18-07930-t001:** Demographic Characteristics of Participants (N = 468).

Variable	No (%)
Age	
15–19	50 (10.68)
20–34	390 (83.33)
≥35	28 (5.98)
Educational status	
Illiterate	349 (74.6)
Primary school	90 (19.2)
Secondary school	29 (6.2)
Wealth tertiles	
Lowest	156 (33.33)
Middle	156 (33.33)
Highest	156 (33.33)
Maternal occupation	
Housewife/Farmer	439 (93.80)
Parity	
Multiparous	382 (81.6)
Household food security status	
Food insecure	276 (58.97)
History of breastfeeding	
Yes	370 (96.8)
Number of ANC visit	
No ANC visit	56 (12.0)
<4 visits	382 (81.6)
≥4 visits	30 (6.40)

Note. The numbers in the table indicate the frequency (%). Abbreviation: ANC, Antenatal Care.

**Table 2 ijerph-18-07930-t002:** Predictors of breastfeeding knowledge using the BFKQ-AO among 468 pregnant rural Ethiopian women participating in breastfeeding education and support intervention (BFESI).

Variable	Unadjusted β (95% CI)	*p*-Value	Adjusted β (95% CI)	*p*-Value
Age				
15–19	Ref		Ref	
20–34	0.16 (−0.18, 49.3)	0.367	−0.19 (−0.76, 0.38)	0.514
≥35	0.26 (−0.23, 75.3)	0.291	−0.11 (−0.78, 0.56)	0.753
Educational status				
Illiterate	Ref		Ref	
Primary school	−0.13 (−0.39, 0.13)	0.335	−0.13 (−0.42, 0.16)	0.389
Secondary school	−0.00 (−0.44, 0.43)	0.987	−0.03 (−0.58, 0.52)	0.903
Maternal occupation				
Housewife/Farmer	Ref		Ref	
Other	0.68 (0.31, 1.06)	**<0.000 ^†^**	0.56 (0.14, 0.97)	**0.009 ***
Wealth tertiles				
Lowest	Ref		Ref	
Middle	−0.19 (−0.44, 0.05)	0.114	−0.05 (−0.33, 0.22)	0.689
Highest	−0.01 (−0.22 0.19)	0.932	−0.07 (−0.34, 0.19)	0.571
Parity				
Primiparous	Ref		Ref	
Multiparous	0.19 (−0.06, 0.45)	0.139	0.69 (−0.49, 1.88)	0.252
ANC visit				
No ANC visit	Ref		Ref	
<4 visits	−0.12 (−0.35, 0.11)	0.324	−0.03 (−0.29, 0.23)	0.813
≥4 visits	0.18 (−0.26, 0.62)	0.386	0.33 (−0.18, 0.85)	0.203

^†^ *p* < 0.01, significant values. * *p* < 0.05, significant values. Note. Data are given as regression coefficients (β) and 95% confidence intervasl. R2 is 0.05. Abbreviations: BFKQ-AO, breastfeeding knowledge questionnaire in Afan Oromo; ANC, antenatal Care.

**Table 3 ijerph-18-07930-t003:** Predictors of attitude towards infant feeding using the IIFAS-AO among 468 pregnant Ethiopian rural women participating in breastfeeding education and support intervention (BFESI).

Variable	Unadjusted β (95% CI)	*p*-Value	Adjusted β (95% CI)	*p*-Value
Age				
15–19	Ref		Ref	
20–34	0.47 (0.18, 0.77)	0.002 *	0.58 (0.25, 0.90)	**0.001 ***
≥35	0.46 (−0.06, 0.98)	0.083	0.56 (−0.02, 1.13)	0.060
Educational status				
Illiterate	Ref		Ref	
Primary school	−0.16 (−0.41, 0.09)	0.207	−0.09 (−0.32, 0.14)	0.433
Secondary school	−0.00 (−0.37, 0.37)	0.994	0.05 (−0.33, 0.42)	0.813
Maternal occupation				
Housewife/Farmer	Ref		Ref	
Other	0.08 (−0.31, 0.46)	0.702	−0.09 (−0.46, 0.26)	0.593
Wealth tertiles				
Lowest	Ref		Ref	
Middle	0.08 (−0.14, 0.30)	0.478	0.14 (−0.08, 0.35)	0.217
Highest	0.15 (−0.08, 0.38)	0.206	0.15 (−0.08, 0.37)	0.203
Parity				
Primiparous	Ref		Ref	
Multiparous	0.03 (−0.18, 0.25)	0.778	−0.25 (−0.48, −0.03)	**0.027 ***
ANC visit				
No ANC visit	Ref		Ref	
<4 visits	0.12 (−0.17, 0.40)	0.430	0.16 (−0.11, 0.43)	0.238
≥4 visits	0.49 (0.12, 0.87)	**0.009 ***	0.41 (0.08, 0.75)	**0.017 ***
BFKQ-AO score	0.33 (0.25, 0.41)	**<0.000 ^†^**	0.33 (0.25, 0.41)	**<0.000 ^†^**

^†^ *p* < 0.01, significant values. *****
*p* < 0.05, significant values. Note. Data are given as regression coefficients (β) and 95% confidence intervals. R2 is 0.15. Abbreviations: BFKQ-AO, breastfeeding knowledge questionnaire in Afan Oromo; IIFAS-AO, Iowa Infant Feeding Attitude Scale in Afan Oromo; ANC, Antenatal Care.

## Data Availability

The data presented in this study are available on request from the corresponding author, due to privacy restrictions.

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
