# Peer review of "Determinants of Knowledge and Attitude towards Breastfeeding in Rural Pregnant Women Using Validated Instruments in Ethiopia"

_ijerph, 2021, doi:10.3390/ijerph18157930_

Round 1
Reviewer 1 Report
Dear Editor, Dear Authors,
The scientific article submitted for evaluation has a great theoretical and practical value. The discussed topic of the article is extremely interesting and important. It was prepared with full precision, adapted to the requirements of the journal. From the very beginning, there is a tendency of the Authors to carefully consider issues relevant to the topic under study. I estimate the theoretical value of the article as very good. Research tools have been well selected and are appropriate to the assumed goals and objectives of the work. The research results were subjected to a thorough, very good and professional statistical analysis. The results of the research were discussed in an interesting way, the analyzes were performed in a highly correct manner, enriched with good statistical analyzes. Graphical preparation of the results in the form of tables enriched the presentation of the results and made them more credible. The authors' discussion of the research was conducted in an interesting and professional way, comparing their research results with the research results of other Authors. It is worth noting that the authors used a rich literature on the subject to write the article. Valuable and relevant research results. I propose to accept this scientific article for publication due to its high scientific value.
Author Response
Authors response: Thank you for your positive assessment of our paper.
Reviewer 2 Report
It's a nice and well written article, which describes the results of a knowledge/attitude study as start of an education and peer support project. The results of the intervention project are published elsewhere. Authors didn't study the breastfeeding practice, because it was the subject of their intervention project later on. Probably peer support and peer behavior are much more important than knowledge. The group of women interviewed is rather homogenous in a particular area of Ethiopia. That's why the findings of the study may not be applicable across other regions in Ethiopia or elsewhere. This weakness is the most important reason that I wonder if it is worthwhile to publish these results to an international journal. If the authors put more attention to the methodology used and less on the results it could be more interesting for other researchers. But they already published the way they adapted the international validated questionnaires to the local context (ref. 32). It’s up to the redaction to decide if you want to publish this well written article, even if the results aren’t spectacular and almost not applicable elsewhere.
Author Response
It's a nice and well written article, which describes the results of a knowledge/attitude study as start of an education and peer support project. The results of the intervention project are published elsewhere.
Authors response: Thank you for your constructive comments on the manuscript. We have addressed your comments below and we feel that the manuscript is greatly improved as a result.
Authors didn't study the breastfeeding practice, because it was the subject of their intervention project later on. Probably peer support and peer behavior are much more important than knowledge. The group of women interviewed is rather homogenous in a particular area of Ethiopia. That's why the findings of the study may not be applicable across other regions in Ethiopia or elsewhere. This weakness is the most important reason that I wonder if it is worthwhile to publish these results to an international journal. If the authors put more attention to the methodology used and less on the results it could be more interesting for other researchers. But they already published the way they adapted the international validated questionnaires to the local context (ref. 32). It’s up to the redaction to decide if you want to publish this well written article, even if the results aren’t spectacular and almost not applicable elsewhere.
Authors response: We agree with the reviewer that peer support and peer behaviour are important to influence women’s breastfeeding behaviour particularly the practical/tangible support which involves helping women solve breastfeeding problems they may face. However, earlier studies suggest that knowledgea,b,c along with attitude towards breastfeedingd,e,f,g,h,I,j was found to be predictors of breastfeeding signifying the need to include it as the outcome of the intervention. Accordingly, we designed the intervention with the assumption that among the four peer support types (informational, emotional, appraisal and practical support), the informational support would improve knowledge about breastfeeding. Further to this, cognizant of the facts attributed to the benefits of using a validated instrumentk, l, findings from the current study can be applied across other regions in Ethiopia considering the context similarity. Regarding, the current article we do believe that it provides important information about the context in which our interventions study was conducted.
aMaonga AR, Mahande MJ, Damian DJ, Msuya SE. Factors affecting exclusive breastfeeding among women in Muheza District Tanga northeastern Tanzania: a mixed method community-based study. Matern Child Health J. 2016;20:77–87.
bMogre V, Dery M, Gaa PK. Knowledge, attitudes and determinants of exclusive breastfeeding practice among Ghanaian rural lactating mothers. Int Breastfeed J. 2016;11:12.
cEgata G, Berhane Y, Worku A. Predictors of non-exclusive breastfeeding at 6 months among rural mothers in east Ethiopia: a community-based analytical cross-sectional study. Int Breastfeed J. 2013;8:8.
dChen C-H, Chi C-S. Maternal intention and actual behavior in infant feeding at one month postpartum. Acta paediatrica taiwanica. 2003;44(3):140-4.
eDungy CI, McInnes RJ, Tappin DM, Wallis AB, Oprescu F. Infant feeding attitudes and knowledge among socioeconomically disadvantaged women in Glasgow. Maternal and child health journal. 2008;12(3):313-22.
fJessri M, Farmer AP, Maximova K, Willows ND, Bell RC, Team AS. Predictors of exclusive breastfeeding: observations from the Alberta pregnancy outcomes and nutrition (APrON) study. BMC pediatrics. 2013;13(1):77.
gMora Adl, Russell DW, Dungy CI, Losch M, Dusdieker L. The Iowa Infant Feeding Attitude Scale: Analysis of Reliability and Validity1. Journal of Applied Social Psychology. 1999;29(11):2362-80.
hScott JA, Binns CW, Oddy WH, Graham KI. Predictors of breastfeeding duration: evidence from a cohort study. Pediatrics. 2006;117(4):e646-e55.
iSonko A, Worku A. Prevalence and predictors of exclusive breastfeeding for the first six months of life among women in Halaba special woreda, Southern Nations, Nationalities and Peoples’ Region/SNNPR/, Ethiopia: a community based cross-sectional study. Archives of Public Health. 2015;73(1):53.
jThulier D, Mercer J. Variables associated with breastfeeding duration. Journal of Obstetric, Gynecologic & Neonatal Nursing. 2009;38(3):259-68.
kChambers J, McInnes R, Hoddinott P, Alder E. A systematic review of measures assessing mothers' knowledge, attitudes, confidence and satisfaction towards breastfeeding. Breastfeeding review: professional publication of the Nursing Mothers' Association of Australia. 2007;15 3:17-25.
lKimberlin CL, Winterstein AG. Validity and reliability of measurement instruments used in research. American journal of health-system pharmacy: AJHP: official journal of the American Society of Health-System Pharmacists. 2008;65(23):2276-84.
Reviewer 3 Report
Thank you for the opportunity to review this manuscript.
Abstract line 32, the word 'and' is missing from between 'knowledge most'
Introduction: Line 66 - this sentence does not quite make sense. Using validated instruments enhances the likelihood that you are measuring constructs as intended.
Line 64 - provide more details about the previous studies in Ethiopia. Recommend bringing these studies back into the discussion - to compare and contrast with your findings.
Line 68 - 70. Suggest making this two sentences to enhance readability.
Line 80-82 also needs revision for readability.
Methods: typo line 92 'ad'
Line 95 - explain 'kebeles' for the international reader. It would also be helpful to make an overall comment about whether this area has adequate health services, as the numbers in line 95-97 have no context for the international reader.
Section 2.3 Women were identified, but then how and where were they approached? At their homes, or in clinics? If 75% are illiterate, how did they provide written consent?
Section 2.4 What training did the nurses receive? This is important to know as they should have been trained to use a standardised interview protocol to minimise bias.
Line 114 - spell out ANC in full, the first time you use it in the text
Section 2.5 You have set up your study as being preferable to previous studies due to the use of validated instruments. However you have not presented evidence that these instruments are valid and reliable in the Ethiopian context. Why did you adopt a questionnaire from Malaysia? Are there similarities with the population there that make the questionnaire applicable in Ethiopia? Are the concepts in the Iowa Infant Feeding scale relevant to our population e.g. breastfeeding in restaurants? This is a significant limitation and needs to be acknowledged in the limitations section of the paper.
Author Response
Abstract line 32, the word 'and' is missing from between 'knowledge most'
Authors response: Thank you, typo now corrected. The corrected sentence (line 32 of the abstract) reads as follows:
“… inadequate knowledge and most women …”
Introduction: Line 66 - this sentence does not quite make sense. Using validated instruments enhances the likelihood that you are measuring constructs as intended.
Authors response: Thank you, we now revised this section to improve clarity (Page 2, lines76-84). In addition, we have now made extensive efforts to improve the clarity of our manuscript. The revised section now read as:
“The strength of the evidence to support interventions depends on the quality and sensitivity of instruments used to measure outcomesa and the quality of a measuring instrument is related to its reliability and validityb. Valid and reliable instruments to measure knowledge and attitude helps policymakers, researchers and practitioners develop strategies on how to promote breastfeedingb by identifying the levels and predictors of maternal knowledge and attitude towards breastfeeding. As such, interventions that aim to improve the psychosocial factors, such as knowledge and attitude, that influence the decision-making process regarding breastfeeding require instruments that are objective, reliable and valid to measure these outcomesb.”
aChambers J, McInnes R, Hoddinott P, Alder E. A systematic review of measures assessing mothers' knowledge, attitudes, confidence and satisfaction towards breastfeeding. Breastfeeding review: professional publication of the Nursing Mothers' Association of Australia. 2007;15 3:17-25.
bKimberlin CL, Winterstein AG. Validity and reliability of measurement instruments used in research. American journal of health-system pharmacy: AJHP : official journal of the American Society of Health-System Pharmacists. 2008;65(23):2276-84.
Line 64 - provide more details about the previous studies in Ethiopia. Recommend bringing these studies back into the discussion - to compare and contrast with your findings.
Authors response: Thank you for this helpful suggestion, which we have now incorporated some details about these studies in the introduction section (Page 2, lines 63-75) and we make note of this in our discussion section.
“In Ethiopia, existing studies tried to recognize the critical role played by maternal knowledge and attitude as predictors of breastfeeding practicesa,b,c,d,e,f,g,h,I,j. Maternal knowledge was identified as a predictor in a few of these studiesc,f,g,h,I,j. In addition, the attitude towards breastfeeding has been assessed but a significant association was not foundb,ef,i. Surprisingly, the effect of maternal knowledge and attitude remain unclear as these studies vary greatly in their depth with the number and the content of the questions used. The knowledge questions are superficial and lack some important domains such as breastmilk expression, management of breast problems, and practical aspects of breastfeeding. Moreover, they have used non-validated tools to report varied factors across different settings. Due to the lack of validated instruments in Ethiopia, there is a dearth of data on a range of factors known to influence maternal knowledge and attitude towards breastfeeding making the generalizability of previous studies on this issue problematic.”
We have also incorporated previous Ethiopian studies in the discussion section on (pages 7-8, lines 251-254). The revised section now reads as:
“In Ethiopia, although extensive research has been carried on knowledge of women about breastfeedingc,f,g,h,I,j, no single study exists which applied a validated instrument and comparing our findings with these studies is unrealistic.”
Further (page 8, lines 263-266) we write:
“In Ethiopia, none of the studies that assessed women’s’ attitude towards breastfeedingb,ef,i used IIFAS and much uncertainty exists about the relationship between maternal attitude and breastfeeding practice wherein our study provides a new insight into the importance of considering neutral attitude.”
aAdugna B, Tadele H, Reta F, Berhan Y. Determinants of exclusive breastfeeding in infants less than six months of age in Hawassa, an urban setting, Ethiopia. International breastfeeding journal. 2017;12:45-.
bAsfaw MM, Argaw MD, Kefene ZK. Factors associated with exclusive breastfeeding practices in Debre Berhan District, Central Ethiopia: a cross sectional community based study. International breastfeeding journal. 2015;10:23-.
cAwoke N, Tekalign T, Lemma T. Predictors of optimal breastfeeding practices in Worabe town, Silte zone, South Ethiopia. PloS one. 2020;15(4):e0232316-e.
dBelachew A, Tewabe T, Asmare A, Hirpo D, Zeleke B, Muche D. Prevalence of exclusive breastfeeding practice and associated factors among mothers having infants less than 6 months old, in Bahir Dar, Northwest, Ethiopia: a community based cross sectional study, 2017. BMC research notes. 2018;11(1):768-.
eBimerew A, Teshome M, Kassa GM. Prevalence of timely breastfeeding initiation and associated factors in Dembecha district, North West Ethiopia: a cross-sectional study. International breastfeeding journal. 2016;11.
fChekol DA, Biks GA, Gelaw YA, Melsew YA. Exclusive breastfeeding and mothers’ employment status in Gondar town, Northwest Ethiopia: a comparative cross-sectional study. International breastfeeding journal. 2017;12(1):27.
gHoche S, Meshesha B, Wakgari N. Sub-Optimal Breastfeeding and Its Associated Factors in Rural Communities of Hula District, Southern Ethiopia: A Cross-Sectional Study. Ethiop J Health Sci. 2018;28(1):49-62.
hKelkay B, Kindalem E, Tagele A, Moges Y. Cessation of Exclusive Breastfeeding and Determining Factors at the University of Gondar Comprehensive Specialized Hospital, Northwest Ethiopia. International journal of pediatrics. 2020;2020:8431953-.
iTsegaye M, Ajema D, Shiferaw S, Yirgu R. Level of exclusive breastfeeding practice in remote and pastoralist community, Aysaita woreda, Afar, Ethiopia. International breastfeeding journal. 2019;14(1):6.
jEgata, G., Berhane, Y. & Worku, A. Predictors of non-exclusive breastfeeding at 6 months among rural mothers in east Ethiopia: a community-based analytical cross-sectional study. Int Breastfeed J 8, 8 (2013). https://doi.org/10.1186/1746-4358-8-8
Line 68 - 70. Suggest making these two sentences to enhance readability.
Authors response: thank you, we now revised the text to give a more explicit description to enhance the readability of the whole manuscript (Page 2, lines 63-75).
The revised section reads as:
“In Ethiopia, existing studies tried to recognize the critical role played by maternal knowledge and attitude as predictors of breastfeeding practicesa,b,c,d,e,f,g,h,I,j, Maternal knowledge was identified as a predictor in a few of these studiesc, f,g,h,I,j. In addition, the attitude towards breastfeeding has been assessed but a significant association was not foundb,ef,i. Surprisingly, the effect of maternal knowledge and attitude remain unclear as these studies vary greatly in their depth with the number and the content of the questions used. The knowledge questions are superficial and lack some important domains such as breastmilk expression, management of breast problems, and practical aspects of breastfeeding. Moreover, they have used non-validated tools to report varied factors across different settings. Due to the lack of validated instruments in Ethiopia, there is a dearth of data on a range of factors known to influence maternal knowledge and attitude towards breastfeeding making the generalizability of previous studies on this issue problematic.”
aAdugna B, Tadele H, Reta F, Berhan Y. Determinants of exclusive breastfeeding in infants less than six months of age in Hawassa, an urban setting, Ethiopia. International breastfeeding journal. 2017;12:45-.
bAsfaw MM, Argaw MD, Kefene ZK. Factors associated with exclusive breastfeeding practices in Debre Berhan District, Central Ethiopia: a cross sectional community based study. International breastfeeding journal. 2015;10:23-.
cAwoke N, Tekalign T, Lemma T. Predictors of optimal breastfeeding practices in Worabe town, Silte zone, South Ethiopia. PloS one. 2020;15(4):e0232316-e.
dBelachew A, Tewabe T, Asmare A, Hirpo D, Zeleke B, Muche D. Prevalence of exclusive breastfeeding practice and associated factors among mothers having infants less than 6 months old, in Bahir Dar, Northwest, Ethiopia: a community based cross sectional study, 2017. BMC research notes. 2018;11(1):768-.
eBimerew A, Teshome M, Kassa GM. Prevalence of timely breastfeeding initiation and associated factors in Dembecha district, North West Ethiopia: a cross-sectional study. International breastfeeding journal. 2016;11.
fChekol DA, Biks GA, Gelaw YA, Melsew YA. Exclusive breastfeeding and mothers’ employment status in Gondar town, Northwest Ethiopia: a comparative cross-sectional study. International breastfeeding journal. 2017;12(1):27.
gHoche S, Meshesha B, Wakgari N. Sub-Optimal Breastfeeding and Its Associated Factors in Rural Communities of Hula District, Southern Ethiopia: A Cross-Sectional Study. Ethiop J Health Sci. 2018;28(1):49-62.
hKelkay B, Kindalem E, Tagele A, Moges Y. Cessation of Exclusive Breastfeeding and Determining Factors at the University of Gondar Comprehensive Specialized Hospital, Northwest Ethiopia. International journal of pediatrics. 2020;2020:8431953-.
iTsegaye M, Ajema D, Shiferaw S, Yirgu R. Level of exclusive breastfeeding practice in remote and pastoralist community, Aysaita woreda, Afar, Ethiopia. International breastfeeding journal. 2019;14(1):6.
jEgata, G., Berhane, Y. & Worku, A. Predictors of non-exclusive breastfeeding at 6 months among rural mothers in east Ethiopia: a community-based analytical cross-sectional study. Int Breastfeed J 8, 8 (2013). https://doi.org/10.1186/1746-4358-8-8
Line 80-82 also needs revision for readability.
Authors response: Thank you for pointing this out. We have now revised the sentence (page 3, lines 103-104) and it reads as follow
“This study will provide new insight into the factors that influence knowledge and attitude towards breastfeeding using instruments that were validated in an Ethiopian context.”
Methods: typo line 92 'ad'
Authors response: Thank you, typo now corrected (page 3, line 114)
We write:
“… from Addis Ababa and 22km from Jimma town”.
Line 95 - explain 'kebeles' for the international reader. It would also be helpful to make an overall comment about whether this area has adequate health services, as the numbers in line 95-97 have no context for the international reader.
Authors response: Thank you for this important consideration. We have included an explanation about kebele i.e., the smallest administrative unit in Ethiopia (page 3, lines 117). We have also now added a description of the overall healthcare service in terms of its adequacy (page 3, line 120-123). The revised section now read as:
“The area is known for its shortage of human resources for health as evidenced by the ratio of 121 health workers to 1000 people i.e., 0.2 which is far below the national target of 1.6 by 2020 a and the international target of 4.5 b indicating the area has inadequate healthcare service.”
aEthiopia F. Health sector transformation plan I. Addis Ababa. 2015.
bWorld Health Organization. Global strategy on human resources for health: workforce 2030. 2016
Section 2.3 Women were identified, but then how and where were they approached? At their homes, or in clinics? If 75% are illiterate, how did they provide written consent?
Authors response: Thank you for bringing attention to the clarity needed on how women were identified and approached including how consent was obtained. We have now revised and added further detail to section 2.3 of the revised manuscript. sentences addressing all concerns raised (page 3, lines 127-136).
We write:
“All pregnant women who were identified from the antenatal care logbook were invited to a meeting at the health post where the nature and purpose of the trial and eligibility criteria were explained including their right to withdraw from the study at any time. They were allowed to ask questions and relevant information provided accordingly. Then according to their literacy status, written informed consent was secured from participants by asking them to provide their fingerprint or signature. To minimize the chance of missing pregnant women, Women Development Army Leaders were involved in the identification of pregnant women who were not registered on the antenatal care logbook through a home visit and consent was obtained following the above procedures.”
Section 2.4 What training did the nurses receive? This is important to know as they should have been trained to use a standardised interview protocol to minimise bias.
Authors response: Thank you, we have now mentioned the type and content of training provided to nurses who participated in the study as data collectors (page 3, lines 141-145). The revised sentence read as:
“Ten BSc nurses who had experience in data collection and fluent in the local language were trained for 2 days on the main objectives of the study and how to standardize the data collection method. All the required supervision was in place during the training session and on the field during data collection by trained supervisors.”
Line 114 - spell out ANC in full, the first time you use it in the text
Authors response: Thank you, we now revised (page 4, line 153)
Section 2.5 You have set up your study as being preferable to previous studies due to the use of validated instruments. However you have not presented evidence that these instruments are valid and reliable in the Ethiopian context. Why did you adopt a questionnaire from Malaysia? Are there similarities with the population there that make the questionnaire applicable in Ethiopia? Are the concepts in the Iowa Infant Feeding scale relevant to our population e.g. breastfeeding in restaurants? This is a significant limitation and needs to be acknowledged in the limitations section of the paper.
Authors response: We appreciate the suggestion and it is very helpful. Accounting for the given suggestions we have now presented our evidence that demonstrates the benefits of using instruments that are validated in the local context. Moreover, we have revised the sentence to give more explicit evidence that the use of the validated instrument is more preferable (Page 2, lines76-84). The revised sentence reads as:
“The strength of the evidence to support interventions depends on the quality and sensitivity of instruments used to measure outcomesa and the quality of a measuring instrument is related to its reliability and validityb. Valid and reliable instruments to measure knowledge and attitude helps policymakers, researchers and practitioners develop strategies on how to promote breastfeeding a by identifying the levels and predictors of maternal knowledge and attitude towards breastfeeding. As such, interventions that aim to improve the psychosocial factors, such as knowledge and attitude, that influence the decision-making process regarding breastfeeding require instruments that are objective, reliable and valid to measure these outcomes a.”
aChambers J, McInnes R, Hoddinott P, Alder E. A systematic review of measures assessing mothers' knowledge, attitudes, confidence and satisfaction towards breastfeeding. Breastfeeding review: professional publication of the Nursing Mothers' Association of Australia. 2007;15 3:17-25.
bKimberlin CL, Winterstein AG. Validity and reliability of measurement instruments used in research. American journal of health-system pharmacy: AJHP: official journal of the American Society of Health-System Pharmacists. 2008;65(23):2276-84.
In addition, we have now revised and included the reasons why we adopted the questionnaire from Malaysia in the introduction section (page 2, line 88-95) it reads
“For the knowledge assessing questionnaire, we did a literature search to find a valid and reliable instrument to assess maternal knowledge about breastfeeding and we found two instruments: the Breastfeeding Knowledge Questionnaire that was developed in America a and a questionnaire assessing knowledge of breastfeeding that was developed in Malaysia b. We preferred to use the latter one as it contains domains of breastfeeding that were not included in the Breastfeeding Knowledge Questionnaire such as colostrum, breastmilk expression, management of breast problem, and practical aspects of breastfeeding.”
aGrossman LK, Harter C, Hasbrouck C. Testing mothers' knowledge of breastfeeding: instrument development and implementation and correlation with infant feeding decision. Journal of Pediatric & Prenatal Nutrition. 1991;2(2):43-63.
bTengku Ismail TA, Sulaiman Z. Reliability and validity of a Malay-version questionnaire assessing knowledge of breastfeeding. The Malaysian journal of medical sciences: MJMS. 2010;17(3):32-9.
In the published article that describes the adaptation and validation procedures of the instruments, we have explained the modifications we made to the Iowa Infant Feeding Attitude scale. One of the modifications we made was to change the concept of breastfeeding in restaurants to public places e.g. wedding places, market places. For further reference, the following link is attached: PMID:32272963.

Round 2
Reviewer 3 Report
Thank you for the time and effort you have put into revising the paper.